# Adolescent relational behaviour and the obesity pandemic: A descriptive study applying social network analysis and machine learning techniques

**Pilar Marqués-Sánchez[1], María Cristina Martínez-Fernández[1], José Alberto Benítez-Andrades[2]\*, Enedina Quiroga-Sánchez[1], María Teresa García-Ordás[3], Natalia Arias-Ramos[1]**

**1** Faculty of Health Sciences, SALBIS Research Group, Campus de Ponferrada, Universidad de León, León, Spain, **2** Department of Electric, SALBIS Research Group, Systems and Automatics Engineering, Universidad de León, León, León, Spain, **3** SECOMUCI Research Group, Escuela de Ingenierías Industrial e Informática, Universidad de León, León, León, Spain

\* jbena@unileon.es

## Abstract

### Aim

To study the existence of subgroups by exploring the similarities between the attributes of the nodes of the groups, in relation to diet and gender and, to analyse the connectivity between groups based on aspects of similarities between them through SNA and artificial intelligence techniques.

### Methods

235 students from 5 different educational centres participate in this study between March and December 2015. Data analysis carried out is divided into two blocks: social network analysis and unsupervised machine learning techniques. As for the social network analysis, the Girvan-Newman technique was applied to find the best number of cohesive groups within each of the friendship networks of the different classes analysed.

### Results

After applying Girvan-Newman in the three classes, the best division into clusters was respectively 2 for classroom A, 7 for classroom B and 6 for classroom C. There are significant differences between the groups and the gender and diet variables. After applying K-means using population diet as an input variable, a K-means clustering of 2 clusters for class A, 3 clusters for class B and 3 clusters for class C is obtained.

### Conclusion

Adolescents form subgroups within their classrooms. Subgroup cohesion is defined by the fact that nodes share similarities in aspects that influence obesity, they share attributes related to food quality and gender. The concept of homophily, related to SNA, justifies our

**Data Availability Statement:** All relevant data are within the paper and its Supporting information files.

**Funding:** The author(s) received no specific funding for this work.

**Competing interests:** The authors have declared that no competing interests exist.

results. Artificial intelligence techniques together with the application of the Girvan-Newman provide robustness to the structural analysis of similarities and cohesion between subgroups.

## Introduction

The World Health Organisation highlights the enormous problem of obesity worldwide. Its data indicates that more than one billion people are obese, with 650 million adults, 340 million adolescents and 39 million children [1]. Prevalence has been on a very worrying upward curve worldwide due to the impact of oncological, endocrine, cardiovascular, etc. diseases [2]. In fact, in the United States, the American Board of Obesity Medicine (ABOM) has been created to alleviate the consequences of this pandemic [3]. The consequences of overweight and obesity have a direct and indirect impact on health, including negative mental health repercussion such as body image dissatisfaction, eating disorders and stress [4]. Obesity in the paediatric population is associated with comorbid cardiometabolic and psychosocial disorders, obesity in adulthood and reduced life expectancy [5]. Obesity in children and adolescents is an issue that demands research attention and discussion. For more than 20 years, research has shown that the school and family approach influences, the natural history of childhood obesity [6]. In this sense, within the complexity of interventions to achieve successful outcomes in childhood obesity prevention, the school setting seems to be the most appropriate [7]. Evidence indicates that approaches to obesity prevention must be comprehensive and need to consider people's environments and a broad social and societal view of socio-economic backgrounds [8]. The literature indicates that the treatment of obesity consists of four stages, firstly prevention with lifestyle interventions. This is followed by structured weight management. Thirdly a comprehensive intervention and finally the use of medical diets, medication, and surgery [9]. The choice of approach varies based on the severity of obesity, the age of the child and the presence of comorbidities related to obesity [10].

Adolescence, the stakeholder group in this work, is a critical period characterized by significant changes in personality and intense emotions, largely influenced by the social dynamics withing the school environment. Social norms derived from these relationaships play a pivotal role in understanding the obesity epidemic, as they contribute to the establishment of social influence mechanisms that connect with obesity-related behaviours [11]. For example, social anxiety generated especially in the school environment (friends, teachers, etc.), could provide explanation on obesity outcomes in young people and adolescents [12].

Social Network Analysis (SNA) is a framework based on analysing the contacts between interacting units in the network [13]. The elements of a social network are the nodes and connections that represent the interaction between a pair of nodes. The specificity of this framework is the importance given to the social structure and not to the individual. That is, in a group or team of people not only the people are connected but also their goals and objectives [14]. This means that the characteristics or behaviour of an individual could depend on the position he or she has in the social structure [14]. Nurses can apply SNA to different phenomena to discover findings not available by traditional methods [15]. Some of the most relevant examples are the studies focused on networks and Binge drinking [16], smoking [17], social exclusion of obese adolescents [18], among others. Evidently, the most recent ones have focused on networks and pandemics, for example on networks and feelings by COVID-19 [19] or networks and contagion among university students [20]. Based on the topic of the current study, research on SNA and obesity has had a relevant impact among academics. One of the

reference works is the contribution of Christakis and Fowler, who described the Contagion Theory of the obesity underlining that the spread of health habits and their consequences could be identified as a process of social contagion through social networks [21]. In this sense, Burt and Janicik described social contagion as the process by which one person is infected by another with respect to an idea or behaviour on the basis of the network in which he or she is embedded [22]. Understanding these factors influencing diffusion would have a broad impact on health education and the behaviour of individuals [23]. Another reference research that adds research evidence is the contribution of Kayla de la Haye et al, who highlighted the importance of properly modelling friendship selection processes in adolescents, as they generate a process of influence related to obesity [24]. However, studies of SNA and obesity that explain group influence mechanisms in a way that provides useful information for network-based interventions are still lacking [25].

Previous studies have shown that the identification of groups and their health behaviours is relevant, and that it would be a key issue to detect subgroups at risk of developing lifestyle-related diseases [26]. We believe that the analysis of groups of adolescents could provide us with key information to understand the mechanism of the spread of obesity. What is now required is an analytical strategy that models the most cohesive groups within classes in relation to weight, in order to simultaneously observe influence processes. To address these processes of social contagion at the group level with respect to obesity in schools, a number of decisions have been made regarding metrics to address these complex networks. On the one hand, the application of Girvan and Newman's model to analyse the structure of highly cohesive groups in the classroom community [27]. The model explains how these groups are linked through looser connections and applies centrality measures to find the boundaries of the community. The second decision has been to include artificial intelligence approaches through unsupervised learning methods. In particular, attributes of the actors have been used to train clustering models, specifically through the K-means technique, thus corroborating that the SNA methodology and artificial intelligence coincide when it comes to generating cohesive communities or subgroups within a network. The use of artificial intelligence and social network analysis techniques is new in the field of health, and there are already studies that are combining both techniques to obtain results that are useful and that solve problems in the field of health [28–30].

Based on the above, our research question is: is there a correlation between the formation of groups in classroom networks and adolescent obesity? To provide and answer, the following aims have been set: (i) To study the existence of subgroups by exploring similarities between the attributes of groups' nodes, in relation to the diet and gender (context of obesity) and, (ii) to analyse the connectivity between groups based on aspects of the similarities between them through SNA and artificial intelligence techniques.

## Methods

### Literature review

Previous research shows that obesity in adolescence is associated with low self-esteem and depression [31] Further studies have shown that obese adolescents are more likely to have overweight friends than their normal-weight peers [32]. Specifically, non-overweight adolescents were more likely to select a non-overweight friend than an overweight friend, while those who were overweight were largely indifferent to the weight status of their friends. In this sense, overweight adolescents have been observed to occupy peripheral positions within their social network, often being isolated. Thus, SNA can be a particularly relevant tool for detecting overweight young people at risk of exclusion [18].

## Design

In order to achieve the proposed objectives, a cross-sectional descriptive cases study was carried out in the educational environment. Specifically, it was carried out with the participation of students in the third and fourth year of compulsory secondary education.

## Setting and sample

A total of 776 students from 5 different educational centers (30 classrooms) were invited to participate in this study. An initial sample of 276 students was obtained, but after excluding individuals classified according to the WHO as "underweight", the sample was finally made up of 235 students belonging to 11 different classrooms. To analyze the data regarding the objectives proposed in this work, three of the most numerous classrooms was selected (n:118). Sample characteristics are shown in Table 1.

## Data collection

The data related to gender, diet and friendship network were collected between March and December 2015 by trained nurses from the SALBIS research group and the Regional Health Service of Castile and Leon (SACYL). It was carried out by means of a paper format questionnaire in the tutoring class.

For the collection of reticular friendship data, the following question was asked 'Using the following list, indicate how much time you spend with your classmates', adapted from the one used by De la Haye et al. [24] ("hang out with the most"). The data obtained were used to construct a *nxn* matrix which is interpreted as follows: for the rows: a nominates b and for the columns, a is nominated by b.

To assess diet, the KIDMED Test of Adherence to the Mediterranean Diet [33] were used. With the KIDMED Test, a total score of between 0 and 12 points can be obtained and the participants can be classified as poor quality of the diet, needs improvement, and optimal quality (the higher the score, the better the quality of the diet).

## Ethical considerations

Ethics approval permission for data collection was sought from the Castille and Leon Education Department and the Spanish Data Protection Agency (CV: BOCYL-D-12032015-6). Since this research is aimed at minors, the signing of the informed consent by the parents/guardians was mandatory, which was written following the recommendations of the Bioethics Committee of the University of Salamanca. Always respecting the availability of the adolescent and under the principle of voluntariness, each student received the document in a sealed envelope to deliver it to her home. The data was stored in an automated file created specifically for this investigation in compliance with the aforementioned Law on the Protection of Personal Data

**Table 1. Sample characteristics.**

|  | Gender | | |
|  | Male N (%) | Female N (%) | Total (%) |
|---|---|---|---|
| Classroom A | 17 (30.4) | 14 (22.6) | 31 (53.0) |
| Classroom B | 22 (39.3) | 25 (40.3) | 47 (79.6) |
| Classroom C | 17 (30.4) | 23 (47.1) | 40 (77.5) |
| Total (%) | 56 (47.5) | 62 (52.5) | 118 (100.0) |

(Organic Law 15/1999, of December 13, on the Protection of Personal Data), and following the recommendations of the Spanish Data Protection Agency. The ownership of the file was public and it was created through the NOTA electronic service, after publication in the Official Gazette of BOCYL, of the most relevant project data. Given the educational context of the population under study, permission was obtained from the Provincial Directorate of Education of León and the General Director of Educational Innovation and Teacher Training, as long as each center consented to its preparation, was consented by the parents and not interfere with the normal functioning of the teaching activity.

## Data analysis

The data analysis in this research was primarily divided into two key segments: social network analysis (SNA) and unsupervised machine learning techniques.

SNA is a theoretical and methodological approach to investigating social relationships by visualizing these connections as networks and nodes [13]. We employed the Girvan-Newman technique, a specific method to detect communities within these social structures, proven to effectively identify highly cohesive subgroups [34,35]. This technique operates by identifying and eliminating edges in the network that, if removed, would cause significant disruption to the network's cohesion.

In applying the Girvan-Newman technique, our objective was to determine the optimal number of cohesive groups within the friendship networks across different classes analyzed. These networks were represented using square matrices of dimensions 31x31 for class A, 47x47 for class B, and 40x40 for class C. The algorithm was employed with a predetermined minimum of 3 and a maximum of 7 groups for calculation. Following the algorithm's application, we derived a metric known as "modularity," denoted by the letter Q. This measure compares the number of internal links within the groups, with higher values suggesting greater cohesion. Subsequently, we examined whether these cohesive subgroups correlated with specific actor attributes within the network, such as gender and diet.

On the other hand, unsupervised machine learning refers to a type of machine learning that seeks out previously undetected patterns in a dataset without pre-existing labels and with minimal human supervision [36,37]. In this study, we utilized unsupervised machine learning techniques, a type of machine learning that seeks out previously undetected patterns in a dataset without pre-existing labels and with minimal human supervision [36,37]. One such technique employed was the K-means algorithm.

K-means is one of the simplest and most widely used clustering algorithms. It's an iterative algorithm that divides a group of n datasets into k non-overlapping subgroups or clusters, where each dataset belongs to the cluster with the nearest mean [38]. It starts by choosing k initial centroids, where k is a user-defined constant representing the number of clusters. Then, it assigns each data point to the nearest centroid, and re-calculates the centroids as the mean of all data points in the cluster. This process repeats until the centroids do not significantly change between iterations or a set number of iterations is reached.

The effectiveness of the K-means algorithm is highly dependent on the initial random assignments of centroids. Different runs of the algorithm may result in different clusters since the algorithm may converge to local minima. To combat this, the algorithm is often run multiple times with different initial centroid assignments.

In our application, we used significant variables detected between the cohesive groups generated and the existing attributes to implement this algorithm. After the groups were calculated using the K-means algorithm, we compared these groups to those derived through the Girvan-

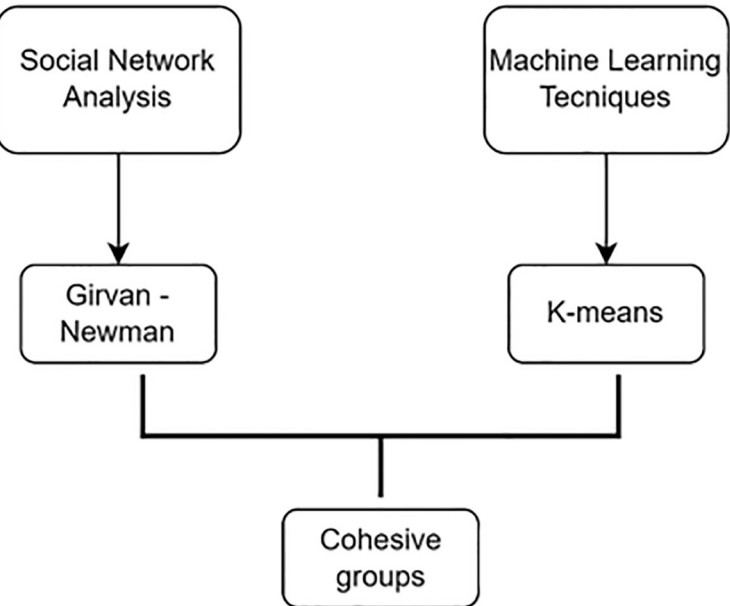

**Fig 1. Diagram of the analysis process.**

Newman technique (see Fig 1). This comparison helped us comprehend the relationship between the cohesive groups obtained using SNA techniques and the groups derived from AI-selected affinity variables.

## Statistical analysis and visualisation

IBM SPSS Statistics (26.0) software was used for the statistical processing of the data. For the analysis of descriptive data, frequencies and percentages were used for the qualitative variables, whereas the mean and standard deviation were used for the quantitative variables. A chi-squared test was carried out to verify whether there was a relationship between the groups. One-factor ANOVA was used to determine whether there were relationships between group membership and the continuous variables. The UCINET tool, version 6.67933 was used for the calculation of the SNA measurements. The tests carried out to study the normality of the distribution were the Shapiro–Wilk test because the sample is smaller than 55. The level of statistical significance was set at 0.05. For qualitative analysis, a visualization of the global network will be carried out using Gephi, version 0.9.3, software.

## Results

### Connectivity between sub-groups

After applying Girvan-Newman in the three classes of the study, the best division into clusters was respectively 2 for classroom A (Q = 0.144), 7 for classroom B (Q = 0.465) and 6 for classroom C (Q = 0.325). Clusters containing less than 3 actors were discarded as they were considered outliers [39]. After the exclusion of outliers, class A remained divided into two groups, unchanged, and classes B and C were divided into 3 groups.

### Attributional characteristics of students in the subgroups in relation to the context of obesity

As shown in Tables 2 and 3, there are significant differences between the groups and the variables of gender and diet. Furthermore, Table 4 shows that among the 2 variables, gender is the most significant variable in all 3 classes.

### SNA vs Artificial Intelligence

After applying K-means using population feeding as an input variable, a K-means clustering of 2 groups for class A, 3 groups for class B and 3 groups for class C is obtained. A comparison is made between the membership of each of these groups in relation to the cohesive groups obtained by Girvan-Newman and the following results are observed:

K-Means groups in class A with Girvan-Newman cohesive groups: a statistically significant similarity is found $\chi^2$ (2, 31) = 5.188, p = 0.023. Within the clusters generated by K-means that we will call KA-1 and KA-2 we find that 90% of the actors belonging to A-1 are in the KA-1 group, while the KA-2 group is composed of 90% of actors from the A-2 group.

K-Means groups in class B with Girvan-Newman cohesive groups: a statistically significant similarity is found $\chi^2$ (4, 47) = 18.166, p = 0.001. Within the clusters generated by K-means that we will call KB-1, KB-2 and KB-3 we find that 60% of the actors in cluster B-1 belong to KB-1, while 70% of B-2 actors belong to KB-2 and 100% of actors in cluster B-3 are distributed between KB-1 and KB-3. Thus, there are no B-1 actors in KB-3, and no B-3 actors in KB-2.

K-Means groups in class C with Girvan-Newman cohesive groups: a statistically significant similarity is found $\chi^2$ (4, 40) = 11.021, p = 0.026. Within the clusters generated by K-means that we will call KC-1, KC-2 and KC-3 we find that 78.6% of the actors in cluster KC-1 belong to C-1, while 66.7% of KC-2 actors belong to C-2 and the actors in cluster KC-3 are distributed between C-1 (56.5%), C-2 (34.8%) and C-3 (8.7%) (Fig 2).

### Discussion

The present study adds evidence to the lack of literature on SNA and children's group behaviour in relation to obesity-related variables. Previous studies have analysed network cohesion as a function of health issues, for example in adolescent sleep [40], well-being [41] or even

**Table 2. Results of the comparison between the different clusters in each class and gender applying the chi-square test of independence.**

|  | Chi square tests of independence | | | |
|  | Male N (%) | Female N (%) | $\chi^2$ | p |
| --- | --- | --- | --- | --- |
| Classroom A |  |  |  |  |
| Cluster A-1 | 3 (27.3) | 8 (72.7) | 5.231 | .022 |
| Cluster A-2 | 14 (70.0) | 6 (30.0) |  |  |
| Classroom B |  |  |  |  |
| Cluster B-1 | 8 (53.3) | 7 (46.7) | 13.263 | .001 |
| Cluster B-2 | 14 (66.7) | 7 (33.3) |  |  |
| Cluster B-3 | 0 (0.0) | 11 (100.0) |  |  |
| Classroom C |  |  |  |  |
| Cluster C-1 | 7 (28.0) | 18 (72.0) | 7.919 | .019 |
| Cluster C-2 | 7 (70.0) | 3 (30.0) |  |  |
| Cluster C-3 | 4 (80.0) | 1 (2.5) |  |  |

**Table 3. Results of the comparison between the different clusters in each class and diet applying the chi-square test of independence.**

| | Chi square tests of independence | | | | |
| --- | --- | --- | --- | --- | --- |
| | Low quality N (%) | Mid N (%) | Best N (%) | $\chi^2$ | $p$ |
| Classroom A | | | | | |
| Cluster A-1 | 0 (0.0) | 10 (90.9) | 1 (10.0) | 5.188 | .023 |
| Cluster A-2 | 0 (0.0) | 10 (50.0) | 10 (50.0) | | |
| Classroom B | | | | | |
| Cluster B-1 | 6 (40.0) | 8 (53.3) | 1 (6.7) | 13.320 | .010 |
| Cluster B-2 | 0 (0.0) | 14 (66.7) | 7 (33.3) | | |
| Cluster B-3 | 4 (36.4) | 3 (27.3) | 4 (36.4) | | |
| Classroom C | | | | | |
| Cluster C-1 | 1 (4.0) | 18 (72.0) | 6 (24.0) | 10.480 | .033 |
| Cluster C-2 | 0 (0.0) | 9 (90.0) | 1 (10.0) | | |
| Cluster C-3 | 2 (40.0) | 3 (60.0) | 0 (0.0) | | |

communication processes during the COVID-19 pandemic [42]. In this study, we found that when we applied Girvan Newman to friendship networks (a strict method to find subgroups within a network), the subgroups found share two characteristics, food quality and gender, with the latter having a statistically higher impact.

According to our results, gender has a very important relevance in the configuration of friendship in adolescence. These results are supported by the concept of homophily included in the SNA paradigm. Homophily is the process by which the degree of connectedness is greater between people who are more similar in culture, behaviour and even genetics [43]. Indeed, studies by Murase et al showed that homophily reinforces local attachment and therefore could develop a segregation effect between social networks [44]. In relation to the study of obesity and its spread through the social network, homophily has been considered as one of the two main causes of this peer contagion [43,45,46], assuming that gender, along with other variables such as race are inherent issues in homophily. Gender seems to be associated with

**Table 4. Logistic regression comparing the effect of sex and diet on membership of the different cohesive subgroups in each of the classes (A, B and C).**

| | Likelihood Ratio Tests | | | |
| --- | --- | --- | --- | --- |
| | Model Fitting Criteria | Chi-Square | df | $p$ |
| Classroom A | | | | |
| Intercept | 12.533 | 4.997 | 1 | .025 |
| Gender | 13.007 | 5.470 | 1 | .019 |
| Diet | 12.406 | 4.869 | 1 | .027 |
| Classroom B | | | | |
| Intercept | 30.027 | 8.616 | 2 | .013 |
| Gender | 39.463 | 18.053 | 2 | < .001 |
| Diet | 31.019 | 9.608 | 2 | .008 |
| Classroom C | | | | |
| Intercept | 20.770 | 1.918 | 2 | .383 |
| Gender | 26.584 | 7.732 | 2 | .021 |
| Diet | 25.513 | 6.661 | 2 | .036 |

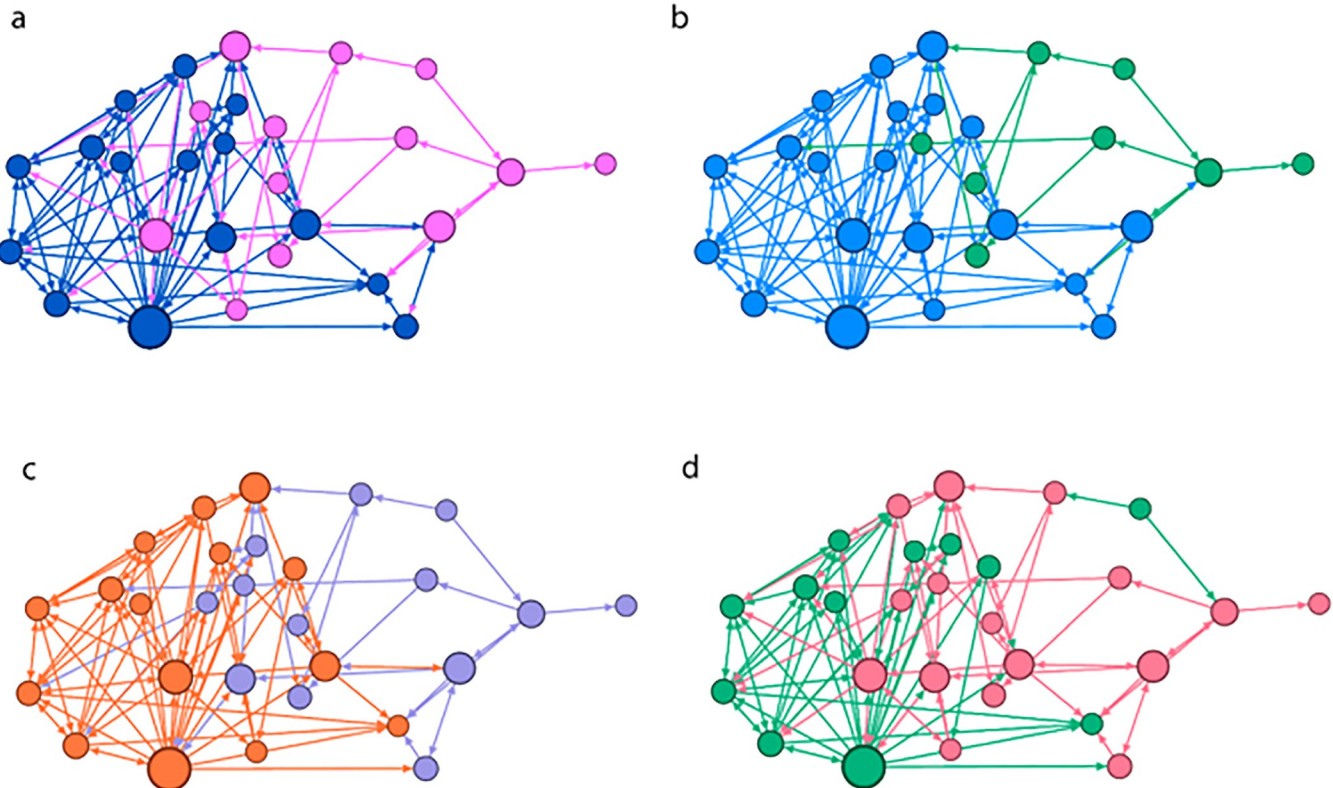

**Fig 2. Representation graphs of the social network formed by class A.** The size represents the degree of intermediation of the nodes and the color represents different attributes in each of the four figures: (a) sex, (b) cohesive subgroups calculated by Girvan-Newman (c) diet and (d) k-means.

negative consequences of obesity. Cheng et al. suggest that obese girls have a lower likelihood of accessing university after high school compared to their non-obese peers, particularly in settings where obesity is less common. Furthermore, this effect was influenced by increased symptoms of internalization, self-medication, and lack of academic commitment. This pattern was not present among males [47]. A review conducted in overweight adolescents suggests that social factors related to weight, such as weight stigma and weight-based discrimination, may be contributing to engagement in risky behaviours [48]. In this regard, and thanks to the SNA, we can identify risk contexts within groups of adolescents. The study of subgroups in health-related issues can offer us the key to carry out networked interventions [49], where value is given to the social context and not only to the characteristics of the individual.

In relation to food habits, this study has shown that most of the subgroups share the same eating habits. Although we cannot establish whether this is due to homophily or influence as the study was not longitudinal, these results are relevant when it comes to developing future interventions in this context. Indeed, routine health promotion in terms of food and nutrition education in school curricula are interventions demanded by health professionals, especially nurses, to combat paediatric obesity [50]. New approaches are required in the prevention and management of obesity, as the literature indicates that lifestyle modifications alone may not be sufficient [10]. Therefore, an approach incorporating SNA could potentially enhance these outcomes. Researchers call for collaboration between institutions to provide comprehensive solutions to obesity, linking environmental, food and school contexts supported by cross-cutting strategies, involving local, regional and state engagement, as well as health and education

policies [51]. This interrelationship of systems, how they communicate and what they transfer, could be analysed within the framework of Social Network Analysis. The literature reports that obesity prevention requires a multidimensional and multisectoral approach that addresses inequality, involves stakeholders and takes into account upstream and downstream factors that influence the risk of obesity [52].

On the other hand, it seems logical to think that it would be easier for an overweight adolescent to share confidences with someone of the same condition in the classroom or on the playground. However, our results show no statistical significance with overweight. This phenomenon could be explained by the overweight adolescents 'strong efforts to integrate into the most cohesive social groups within the class, where both overweight and normal-weight students interact. In fact, De la Haye et al showed that, although overweight students feel more comfortable with similar peers, they would like to share friendships with non-overweight peers, as this option would help them to improve their social status [24]. There was also no statistical significance found between subgroups and physical activity in this study, a variable with much support in the literature. The authors believe that this result may be explained by the influence of parents on physical activity. At this age, adolescents do not engage in physical activity on their own. The practices might be motivated in their own family and welcomed by the adolescents [53]. A longitudinal study conducted a cluster analysis in children and found that those who watched television and consumed unhealthy foods and beverages were more likely to develop obesity. This clusters remained consistent to moderate over a period of three years [54].

In addition to what has been described through the analysis of social networks, it has been possible to verify that the calculation of groups by means of clustering techniques through artificial intelligence, using gender and food as input attributes, generates groups significantly similar to those obtained through Girvan-Newman. This finding demonstrates a direct relationship between the calculation of cohesive subgroups using Girvan-Newman and the generation of subgroups based on common attributes among actors in the network. This result further confirms findings from other studies, albeit on different subjects, supporting the outcomes of our research [28,30].

## Limitations

This study adds scientific evidence on how aspects of similarity and homophily influence health-related behaviours, as could be the case in a pandemic as worrying as obesity. Another strength of this study is the application of SNA and artificial intelligence techniques, which supports the importance of interdisciplinary health and systems engineering teams to develop interventions applied to community health.

However, the authors of this work recognize a series of limitations. The most important limitations are the sample size of each network to be studied and also the analysis of other attributes of the nodes that could enrich the information provided with respect to the formation of the subgroups. We also believe that asking more relational questions can improve the understanding of the subgroup.

As future research fields, we propose to apply the model of this study to other relational contexts such as alcohol and cannabis use among adolescents. The application of SNA to relational contexts allows us to delve deeper into how behaviours respond to social structure, a key point that is not addressed by other methodologies applied to health sciences.

## Conclusion

The present research study has analysed the relational environment between students and obesity-related data. The objectives addressed in this study were twofold: 1) analyzing the

similarities between the adolescents' attributes in relation to diet and gender, and 2) examining the connectivity between groups based on their similarities using SNA with Girvan-Newman and artificial intelligence techniques. The main conclusions are listed below:

- Teenagers form subgroups or sub-networks within their school classrooms.

- Subgroup cohesion is defined by the fact that nodes share similarities in aspects that influence obesity.

- The subgroups were found to share attributes related to food quality and gender.

- The concept of homophily related to SNA, justifies our results.

- Artificial intelligence techniques together with the application of the Girvan-Newman SNA metric offer robustness to the structural analysis of similarities and cohesion between subgroups.

## Supporting information

**S1 Data.**
(SAV)

## Author Contributions

**Conceptualization:** Pilar Marqués-Sánchez, María Cristina Martínez-Fernández, José Alberto Benítez-Andrades, Enedina Quiroga-Sánchez, María Teresa García-Ordás, Natalia Arias-Ramos.

**Data curation:** Pilar Marqués-Sánchez, María Cristina Martínez-Fernández, José Alberto Benítez-Andrades, Enedina Quiroga-Sánchez, María Teresa García-Ordás, Natalia Arias-Ramos.

**Formal analysis:** Pilar Marqués-Sánchez, María Cristina Martínez-Fernández, José Alberto Benítez-Andrades, Enedina Quiroga-Sánchez, María Teresa García-Ordás, Natalia Arias-Ramos.

**Investigation:** José Alberto Benítez-Andrades, Enedina Quiroga-Sánchez, María Teresa García-Ordás, Natalia Arias-Ramos.

**Methodology:** María Cristina Martínez-Fernández, José Alberto Benítez-Andrades, Enedina Quiroga-Sánchez, María Teresa García-Ordás, Natalia Arias-Ramos.

**Project administration:** Pilar Marqués-Sánchez, José Alberto Benítez-Andrades, Natalia Arias-Ramos.

**Resources:** Pilar Marqués-Sánchez, José Alberto Benítez-Andrades, Enedina Quiroga-Sánchez, María Teresa García-Ordás, Natalia Arias-Ramos.

**Software:** Pilar Marqués-Sánchez, María Cristina Martínez-Fernández, José Alberto Benítez-Andrades, Enedina Quiroga-Sánchez, María Teresa García-Ordás, Natalia Arias-Ramos.

**Supervision:** Pilar Marqués-Sánchez, José Alberto Benítez-Andrades, María Teresa García-Ordás, Natalia Arias-Ramos.

**Validation:** Pilar Marqués-Sánchez, José Alberto Benítez-Andrades, Enedina Quiroga-Sánchez, María Teresa García-Ordás, Natalia Arias-Ramos.

**Visualization:** Pilar Marqués-Sánchez, María Cristina Martínez-Fernández, José Alberto Benítez-Andrades, María Teresa García-Ordás, Natalia Arias-Ramos.

**Writing – original draft:** Pilar Marqués-Sánchez, María Cristina Martínez-Fernández, José Alberto Benítez-Andrades, Enedina Quiroga-Sánchez, Natalia Arias-Ramos.

**Writing – review & editing:** Pilar Marqués-Sánchez, María Cristina Martínez-Fernández, José Alberto Benítez-Andrades, Enedina Quiroga-Sánchez, María Teresa García-Ordás, Natalia Arias-Ramos.

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
