## [Decision Letter · Decision Letter 0]

22 May 2023

PONE-D-23-04755ADOLESCENT RELATIONAL BEHAVIOUR AND THE OBESITY PANDEMIC: A DESCRIPTIVE STUDY APPLYING SOCIAL NETWORK ANALYSIS AND MACHINE LEARNING TECHNIQUESPLOS ONE

Dear Dr. Benítez-Andrades,

Thank you for submitting your manuscript to PLOS ONE. After careful consideration, we feel that it has merit but does not fully meet PLOS ONE’s publication criteria as it currently stands. Therefore, we invite you to submit a revised version of the manuscript that addresses the points raised during the review process.

We look forward to receiving your revised manuscript.

Kind regards,

Ricardo Limongi

Academic Editor

PLOS ONE

Journal Requirements:

Additional Editor Comments:

Dear author,

A decision was made on the article submitted to the Pretext Journal: "ADOLESCENT RELATIONAL BEHAVIOUR AND THE OBESITY PANDEMIC: A DESCRIPTIVE STUDY APPLYING SOCIAL NETWORK ANALYSIS AND MACHINE LEARNING TECHNIQUES".

The decision is major revision allowing improvements from the contribution of the reviewers.

According to the consolidated opinion made available below, we emphasize that the new version should contemplate a clearer and more robust discussion about the theoretical position adopted; argumentation for the chosen analysis criteria and statistical interpretation; and, better discussion with the findings of the literature on the subject.

We greatly appreciate your interest in choosing PLOS ONE Magazine as a means for the dissemination of your work.

Thank you very much

-=-=-=-=-=-=-=-=-=-=-=-=-=-=-=-=-=-=-=-=-=-=-=-=-=-=-=-=-=-=-=-=-=-=-=-=-=-=-=-=-=-=-=-=-=-=-=-=-=-=-=-=-=-=-=-=-=-=-=-=-=-=-=-=-=-=-=-=-=-=-=-=-=-=-=-=-=-=-=-=-=-=-=-=-=-=-=-=-=-=-=-=-=-=-=-=-=-=-=-=-=-=-=-=-=-=-=-=-=-=-=-=-=-=-=-=-=-=-=

The manuscript entitled "ADOLESCENT RELATIONAL BEHAVIOUR AND THE OBESITY PANDEMIC: A DESCRIPTIVE STUDY APPLYING SOCIAL NETWORK ANALYSIS AND MACHINE LEARNING TECHNIQUES" aimed to study the existence of subgroups by exploring the similarities between the attributes of the nodes of the groups in relation to diet and gender and, to analyze the connectivity between groups based on aspects of similarities between them through SNA and artificial intelligence techniques and from a Quantitative approach has as main conclusion that adolescents form subgroups in classrooms based on shared attributes related to obesity, food quality, and gender, demonstrating subgroup cohesion. Homophily, a concept in social network analysis, supports these findings. Applying artificial intelligence techniques and the Girvan-Newman algorithm enhances the analysis of similarities and cohesion within these subgroups.

As a way to contribute to the advancement of the manuscript, considerations will be presented according to the respective sections:

Introduction

1. For a better detailing of the contribution, relevance, and theoretical cut, the introduction could be organized, by means of paragraphs, as follows: (i) Common ground (what we know in the literature): The author must present the basic premises of the literature under analysis, to establish a starting point for the reader. (ii) Complication (what are the limitations – gaps – of the literature): The author must expose missing elements or flaws in the literature (incoherent, incomplete, or contradictory theories) to explain some phenomenon that it should explain. Practical, real-world examples can help expose the limitations of the literature. (iii) Concern: The author must show that the limitation of the literature is something relevant to be studied. A trivial limitation is insufficient to motivate a search. The author should write How and Why limitation impairs our ability to understand a phenomenon. (iv) Course of Action: The author should show the course of action that may resolve the central complication. To resolve the limitation, the author must develop a theory or refine an existing theory. To this end, the author can (i) suggest new constructs; (ii) model new relationships between constructs; (iii) explore a theoretical process; (iv) develop a typology. (v) Contribution: The author must explain how his course of action can contribute significantly to the existing theory (that which has been described on the common ground).

Method

1. The authors could briefly discuss how previous studies have investigated the perspective of behavior in relation to the obesity pandemic. Thus, the reader will have greater clarity in understanding the evolution and relevance of the topic. The discussion could be inserted into a section titled "Literature Review.”

2. For better replicability of the study, the authors could present a flowchart of the analytical proposal.

3. Each criterion of choice must compose a justification so the reader understands the methodological paths.

4. It could present the indicators and algorithms used in the cluster analysis to clarify the results.

Data Analysis

1. The analysis in an exploratory perspective and then by mean test, and then logistic is methodologically coherent; however, with the absence of detail of the methodological procedures, it is not clear the evolution followed to evaluate the proposed objective.

Reviewers' comments:

Reviewer's Responses to Questions

**Comments to the Author**

1. Is the manuscript technically sound, and do the data support the conclusions?

Reviewer #1: Yes

2. Has the statistical analysis been performed appropriately and rigorously? 

Reviewer #1: I Don't Know

3. Have the authors made all data underlying the findings in their manuscript fully available?

Reviewer #1: No

4. Is the manuscript presented in an intelligible fashion and written in standard English?

Reviewer #1: Yes

5. Review Comments to the Author

Reviewer #1: Without considering the novelty, overall, the article provides a clear overview of the importance of studying obesity in different groups of people, as well as the potential benefits of using social network analysis (SNA) to understand the mechanisms of obesity spread in schools. However, here are a few suggestions for improvement:

1. Provide more context: While the article provides a good introduction to the issue of obesity in children and adolescents, it could benefit from additional background information on the topic. For example, it might be helpful to briefly discuss the causes and consequences of obesity, as well as some of the current strategies being used to address the problem.

2. Clarify the research questions: The article outlines the overall objective of the study, but it could benefit from more specific research questions to guide the research.

3. Provide more detail on the methodology: While the article discusses the use of SNA and artificial intelligence, it is suggested to add some more background knowledge regarding the techniques used in this research.

4. Highlight the potential impact: While the article mentions the potential impact of the research on health education and behavior change, it is suggested to add more explicit discussion of the potential impact of the findings.

5. Consider the readability: While the article is generally well-written, there are some sentences that are quite long and complex. Consider breaking them up into smaller sentences to improve readability.

6. PLOS authors have the option to publish the peer review history of their article (what does this mean?). If published, this will include your full peer review and any attached files.

Reviewer #1: **Yes: **Kai Ding

---

## [Author Response · Author response to Decision Letter 0]

28 Jun 2023

Rebuttal letter

Q1: 

The manuscript entitled "ADOLESCENT RELATIONAL BEHAVIOUR AND THE OBESITY PANDEMIC: A DESCRIPTIVE STUDY APPLYING SOCIAL NETWORK ANALYSIS AND MACHINE LEARNING TECHNIQUES" aimed to study the existence of subgroups by exploring the similarities between the attributes of the nodes of the groups in relation to diet and gender and, to analyze the connectivity between groups based on aspects of similarities between them through SNA and artificial intelligence techniques and from a Quantitative approach has as main conclusion that adolescents form subgroups in classrooms based on shared attributes related to obesity, food quality, and gender, demonstrating subgroup cohesion. Homophily, a concept in social network analysis, supports these findings. Applying artificial intelligence techniques and the Girvan-Newman algorithm enhances the analysis of similarities and cohesion within these subgroups.

AQ1:

Thank you for taking the time to review our manuscript entitled "ADOLESCENT RELATIONAL BEHAVIOUR AND THE OBESITY PANDEMIC: A DESCRIPTIVE STUDY APPLYING SOCIAL NETWORK ANALYSIS AND MACHINE LEARNING TECHNIQUES" We sincerely appreciate your insightful comments and suggestions, which have greatly contributed to improving the quality and clarity of our study.

In response to your feedback, we have carefully considered the respective sections and have made revisions accordingly.

Q2:

As a way to contribute to the advancement of the manuscript, considerations will be presented according to the respective sections:

Introduction

1. For a better detailing of the contribution, relevance, and theoretical cut, the introduction could be organized, by means of paragraphs, as follows: (i) Common ground (what we know in the literature): The author must present the basic premises of the literature under analysis, to establish a starting point for the reader. (ii) Complication (what are the limitations – gaps – of the literature): The author must expose missing elements or flaws in the literature (incoherent, incomplete, or contradictory theories) to explain some phenomenon that it should explain. Practical, real-world examples can help expose the limitations of the literature. (iii) Concern: The author must show that the limitation of the literature is something relevant to be studied. A trivial limitation is insufficient to motivate a search. The author should write How and Why limitation impairs our ability to understand a phenomenon. (iv) Course of Action: The author should show the course of action that may resolve the central complication. To resolve the limitation, the author must develop a theory or refine an existing theory. To this end, the author can (i) suggest new constructs; (ii) model new relationships between constructs; (iii) explore a theoretical process; (iv) develop a typology. (v) Contribution: The author must explain how his course of action can contribute significantly to the existing theory (that which has been described on the common ground).

AQ2:

Thank you for your comment. We have carefully considered your feedback and have restructured the information according to the guidelines you provided. Following your guidance, we have organized the introduction into paragraphs that address the key aspects you mentioned.

Q3:

Method

1. The authors could briefly discuss how previous studies have investigated the perspective of behavior in relation to the obesity pandemic. Thus, the reader will have greater clarity in understanding the evolution and relevance of the topic. The discussion could be inserted into a section titled "Literature Review.”

AQ3:

Thank you for suggesting a literature review section to analyse previous studies on behaviour and the obesity pandemic. We have incorporated this into the methodology, as you suggest.

Lines 132-140.

Q4:

2. For better replicability of the study, the authors could present a flowchart of the analytical proposal.

AQ4:

Thank you for your comment. To better understand the analytical proposal a flowchart has been added (Figure 1). 

Q5:

3. Each criterion of choice must compose a justification so the reader understands the methodological paths.

AQ5:

We appreciate the reviewer for their insightful comment. Indeed, it is necessary to expand on this information for a proper understanding of the procedure and replicability. Therefore, taking into account their suggestions, the data analysis section has been rewritten.

Q6:

4. It could present the indicators and algorithms used in the cluster analysis to clarify the results.

Data Analysis

1. The analysis in an exploratory perspective and then by mean test, and then logistic is methodologically coherent; however, with the absence of detail of the methodological procedures, it is not clear the evolution followed to evaluate the proposed objective.

AQ6:

We appreciate your suggestion. In accordance with your comments, we have provided more details in this section.

Reviewers' comments:

Reviewer's Responses to Questions

Comments to the Author

Reviewer #1

Q7:

Without considering the novelty, overall, the article provides a clear overview of the importance of studying obesity in different groups of people, as well as the potential benefits of using social network analysis (SNA) to understand the mechanisms of obesity spread in schools. However, here are a few suggestions for improvement:

AQ7:

The authors are grateful for the efforts taken by the reviewer in reading and providing feedback on our manuscript. We appreciate all their comments and hope that our responses are adequate to their suggestions.

Q8:

1. Provide more context: While the article provides a good introduction to the issue of obesity in children and adolescents, it could benefit from additional background information on the topic. For example, it might be helpful to briefly discuss the causes and consequences of obesity, as well as some of the current strategies being used to address the problem.

AQ8:

Thank you for your comment. We agree that additional background information would enhance the overall understanding of the topic. 

In response to your comment, we have included a section in the introduction that briefly summarizes those issues:

“The consequences of overweight and obesity have a direct and indirect impact on health, including negative mental health repercussion such as body image dissatisfaction, eating disorders and stress (4). Obesity in the paediatric population is associated with comorbid cardiometabolic and psychosocial disorders, obesity in adulthood and reduced life expectancy (5).”

“Evidence indicates that approaches to obesity prevention must be comprehensive and need to consider people’s environments and a broad social and societal view of socio-economic backgrounds (8). The literature indicates that the treatment of obesity consists of four stages, firstly prevention with lifestyle interventions. This is followed by structured weight management. Thirdly a comprehensive intervention and finally the use of medical diets, medication, and surgery (9). The choice of approach varies based on the severity of obesity, the age of the child and the presence of comorbidities related to obesity (10).”

Q9:

2. Clarify the research questions: The article outlines the overall objective of the study, but it could benefit from more specific research questions to guide the research.

AQ9:

Thank you for your suggestion. We have revised the manuscript and included a well-defined research question that aligns with the objectives of our study. 

“Based on the above, our research question is: is there a correlation between the formation of groups in classroom networks and adolescent obesity? To provide and answer, the following aims have been set: (i) To study the existence of subgroups by exploring similarities between the attributes of groups' nodes, in relation to the diet and gender (context of obesity) and, (ii) to analyse the connectivity between groups based on aspects of the similarities between them through SNA and artificial intelligence techniques.”

Q10:

3. Provide more detail on the methodology: While the article discusses the use of SNA and artificial intelligence, it is suggested to add some more background knowledge regarding the techniques used in this research.

AQ10:

We are grateful for your suggestion and, in response to it, we have made major changes to the "Data Analysis" section. We hope that it now contains the necessary information.

Q11:

4. Highlight the potential impact: While the article mentions the potential impact of the research on health education and behavior change, it is suggested to add more explicit discussion of the potential impact of the findings.

AQ11:

Thank you for your comment. We have included additional studies that highlight the implications and future utility of our work in the sections Discussion and Conclusions. 

We believe these additions strengthen the article and address the reviewer's comment. Thank you for bringing this to our attention and allowing us to enhance the clarity and significance of our research.

Q12:

5. Consider the readability: While the article is generally well-written, there are some sentences that are quite long and complex. Consider breaking them up into smaller sentences to improve readability.

AQ12:

We thank the reviewer for his comment. We have made drafting improvements throughout the document to make it more readable.

---

## [Editor Report · Decision Letter 1]

21 Jul 2023

ADOLESCENT RELATIONAL BEHAVIOUR AND THE OBESITY PANDEMIC: A DESCRIPTIVE STUDY APPLYING SOCIAL NETWORK ANALYSIS AND MACHINE LEARNING TECHNIQUES

PONE-D-23-04755R1

Dear Dr. Benítez-Andrades,

We’re pleased to inform you that your manuscript has been judged scientifically suitable for publication and will be formally accepted for publication once it meets all outstanding technical requirements.

Kind regards,

Ricardo Limongi

Academic Editor

PLOS ONE

---

## [Editor Report · Acceptance letter]

2 Aug 2023

PONE-D-23-04755R1 

Adolescent Relational Behaviour and the Obesity Pandemic: A descriptive study applying social network analysis and machine learning techniques 

Dear Dr. Benítez-Andrades:

I'm pleased to inform you that your manuscript has been deemed suitable for publication in PLOS ONE. Congratulations! Your manuscript is now with our production department. 

Kind regards, 

on behalf of

Professor Ricardo Limongi 

Academic Editor

PLOS ONE